# Mapping vaccination coverage to explore the effects of delivery mechanisms and inform vaccination strategies

C. Edson Utazi [1,2], Julia Thorley[1], Victor A. Alegana[1,3], Matthew J. Ferrari[4], Saki Takahashi [5], C. Jessica E. Metcalf [5], Justin Lessler[6], Felicity T. Cutts[7] & Andrew J. Tatem[1,3]

The success of vaccination programs depends largely on the mechanisms used in vaccine delivery. National immunization programs offer childhood vaccines through fixed and out-reach services within the health system and often, additional supplementary immunization activities (SIAs) are undertaken to fill gaps and boost coverage. Here, we map predicted coverage at 1×1 km spatial resolution in five low- and middle-income countries to identify areas that are under-vaccinated via each delivery method using Demographic and Health Surveys data. We compare estimates of the coverage of the third dose of diphtheria-tetanus-pertussis-containing vaccine (DTP3), which is typically delivered through routine immunization (RI), with those of measles-containing vaccine (MCV) for which SIAs are also undertaken. We find that SIAs have boosted MCV coverage in some places, but not in others, particularly where RI had been deficient, as depicted by DTP coverage. The modelling approaches outlined here can help to guide geographical prioritization and strategy design.

[1] WorldPop, School of Geography and Environmental Science, University of Southampton, Southampton SO17 1BJ, UK. [2] Southampton Statistical Sciences Research Institute, University of Southampton, Southampton SO17 1BJ, UK. [3] Flowminder Foundation, Stockholm SE- 11355, Sweden. [4] Center for Infectious Disease Dynamics, The Pennsylvania State University, State College, PA 16802, USA. [5] Department of Ecology and Evolutionary Biology, Princeton University, Princeton, NJ 08544, USA. [6] Department of Epidemiology, Johns Hopkins Bloomberg School of Public Health, Baltimore, MD 21205, USA. [7] Department of Infectious Disease Epidemiology, London School of Hygiene and Tropical Medicine, London WC1E 7HT, UK. Correspondence and requests for materials should be addressed to C.E.U. (email: c.e.utazi@soton.ac.uk)

The coverage of childhood vaccines—the proportion of the target population that is vaccinated—is an important indicator of the performance of national health and immunization systems. The recent increasing call for subnational health metrics as part of monitoring equity has generated substantial interest in producing high-resolution maps of vaccination coverage and the associated uncertainties[1,2] to support program planning and implementation. Geographical inequities in vaccination coverage, among other factors, are greatly influenced by the delivery mechanisms adopted by countries in the administration of vaccines[3]. While some delivery mechanisms are universal across all vaccine types, other strategies used in vaccine delivery are antigen-specific[3,4]. Understanding the spatial variation in vaccination coverage in the context of varying vaccine delivery strategies is important for evaluating the performance of vaccination programs.

Since the establishment of the World Health Organisation's Expanded Programme on Immunization (WHO EPI) in 1974 (ref. [5]), the core vaccine delivery mechanism is routine immunization (RI)[6,7], a critical component of the health system that ensures regular provision of immunization services to target populations, following the national vaccination schedule. In some countries, RI systems incorporate tailored activities such as periodic intensification of routine immunization or child health days[7], designed to achieve short-term improvements in RI coverage. Mass campaigns known as supplementary immunization activities (SIAs) involve the delivery of certain vaccines (e.g. oral polio vaccine, measles-containing vaccine (MCV)) to target populations irrespective of their previous vaccination status over a short, planned period of time[8–10]. Conceptually, SIAs focus on interrupting disease circulation and generation of herd immunity and thus play a key role in disease control and elimination efforts[10–13]. However, periodic SIAs have become a mechanism for boosting immunization coverage and filling immunity gaps in many low- and middle-income countries (LMICs) with underperforming health systems[14].

Despite the long history of RI and SIAs in control and elimination programs, little is known about spatial variation in their relative effectiveness in many countries. Identifying areas where RI is lagging is key for planning targeted strategies to improve system performance. SIAs are often reported to have high coverage levels[13], but it is unclear whether they have been equally effective in reaching children in all areas, especially where RI is weak. Here, we explore vaccination coverage at fine spatial scales across five LMICs (Nigeria, Ethiopia, the Democratic Republic of Congo (DRC), Cambodia and Mozambique) to assess differences between delivery mechanisms, using Demographic and Health Surveys (DHS) data. Specifically, we compare the coverage of the third dose of diphtheria–tetanus–pertussis-containing vaccine (DTP3), which is administered through RI[6,15], with the coverage of at least the first dose of MCV, for which SIAs are often undertaken[14,16]. Furthermore, we use high-resolution maps of the coverage of each dose of DTP-containing vaccine (DTP1–3) and the dropout rates between these doses (see, for example, ref. [15]) to assess geographical variation in access to and acceptance of RI (indicated by DTP1 coverage)[6,17] and in health system continuity (i.e. the ability of health systems to follow-up with subsequent doses). These maps are also integrated with high-resolution population maps to produce estimates of numbers of under-vaccinated children which are important to assess outbreak risk, and to evaluate progress towards coverage targets.

## Results

**Mapping DTP1–3 and MCV coverage.** Selected covariates for mapping vaccination coverage for each country and vaccination type are presented in Supplementary Tables 1 and 2. There were strong similarities between the selected covariate sets for MCV and DTP1–3 especially in Nigeria and DRC, with remoteness related variables selected in all cases.

Supplementary Tables 3–7 report the estimates of the regression parameters of the fitted models as well as the parameters of the spatial random effects. For MCV, covariates with statistically significant association (i.e. covariates for which the coefficients had 95% credible intervals—the intervals formed by the 2.5% and 97.5% quantiles—that did not include zero) with vaccination coverage were: poverty (Nigeria), travel time to major cities (Nigeria, Mozambique and Cambodia), population density (Cambodia and Ethiopia), night-time light (Nigeria), distance to urban areas (Ethiopia, DRC), distance to infrastructures (Cambodia), distance to water (Ethiopia), livestock densities (DRC) and environmental variables (Nigeria, DRC and Mozambique). Similar sets of covariates also showed strong associations with all the three doses of DTP in Nigeria (travel time and aridity), Ethiopia (distance to urban areas, travel time, livestock density), DRC (distance to urban areas, environmental variables and livestock density), Cambodia (distance to populated places and environmental variables) and Mozambique (distance to highways). In all cases, remoteness as measured by travel time to the nearest major settlement and the distance variables consistently had a significant negative association with coverage of both vaccines, matching findings elsewhere[18].

The parameters of the spatial random effects are interpreted as follows. The estimates of the decay parameters and variance/covariance parameters for MCV and DTP1–3 are all significant, suggesting that the covariates included in the models did not account for all of the spatial correlation in vaccination coverage. For MCV, the effective ranges for Nigeria, Ethiopia and DRC (between 350 and 392 km) were relatively longer compared to Cambodia and Mozambique (<224 km), although these estimates are also dependent upon the geographic extents of these countries and the prior distributions assigned to these parameters. For DTP, the effective ranges for Nigeria were 385, 761 and 588 km for doses 1, 2 and 3, respectively. For all other countries, the effective range for DTP1 (274–1894 km) was consistently greater than those of DTP2 (61–896 km) and DTP3 (68–1565 km). The cross-covariance parameters show that significant spatial interdependence exists among the three DTP doses (conditional on the covariates included in the models). Correlations of at least 0.91 were estimated between the doses through the underlying spatial processes.

Table 1 provides the predictive performance statistics for the fitted models. The percentage bias estimates of the fitted models are generally low, ranging between −1.47% and 0.54% for MCV, and between −2.23% and 0.44% for DTP1–3. The reported values indicate that the fitted models are more likely to slightly underestimate MCV and DTP1–3 coverage in some areas especially in Nigeria, Cambodia, Mozambique and Ethiopia, whereas the opposite is the case in DRC. The validation mean square errors (VMSE) of all the models are close to zero in most cases, demonstrating further that the predictions from the models are close to their true values and have small bias. The actual coverage of the 95% prediction intervals measures the accuracy of the uncertainties associated with the predictions. For Cambodia, the achieved coverages of the 95% prediction intervals were close to the true value of 95% for both MCV and all three DTP doses. For Nigeria, Mozambique and Ethiopia, the achieved coverages fall between 80.43% and 88.67%. However, for DRC (both MCV and DTP1–3), the prediction intervals were narrower resulting in poorer coverage levels of between 63.14% and 72.15%. This indicates that the uncertainties associated with the distributions of the predictions were less accurate in these cases, although their

central tendencies as measured by the VMSE and %bias were plausible. Concerning the predictive power of the fitted models for MCV, moderate to high $R^2$ values (≥0.50, calculated as the square of the sample correlation coefficient between observed and predicted values) were obtained in all the countries, except for Cambodia which had a slightly lower value of 0.42. This suggests that, conditional on the spatial random effects, the covariates included in the fitted models accounted for a significant amount of variation in the observed vaccination coverage. For DTP1–3, the $R^2$ values were at least 0.51, except for Cambodia with $R^2$ values ranging between 0.30 and 0.38. In general, these statistics show that the fitted models performed well in most cases (given the difficulties often associated with predicting binomial probabilities[2,19]).

The predicted coverage maps for MCV and DTP1–3 and the corresponding standard deviation (uncertainty) maps are displayed in Fig. 1 and Supplementary Figs. 1–5 for all five countries. The standard deviation maps show areas where predictions were made with high confidence (low s.d.) and areas where confidence is lower (high s.d.). Uncertainty levels were generally low (s.d. < 0.28 for DTP1–3 and s.d. < 0.20 for MCV) for both vaccinations across all five countries. We discuss the patterns in the predicted maps below.

**Comparing DTP1–3 coverage maps.** There is a considerable amount of variation in the coverage levels of the three doses of DTP within countries, especially in DRC, Nigeria and Ethiopia (Fig. 1, Supplementary Figs. 2–5) where marked geographical patterns emerged. Using DRC as a representative case, Fig. 1 shows that in 2014, consistently high coverage areas were located in Bas-Congo and neighbouring Kinshasa, the eastern areas of Nord- and Sud-Kivu and parts of the southern axes, whereas consistently poor coverage areas occurred in the north-central areas, cutting across parts of Maniema, Kasai-Oriental and the province of Tanganyika. Following the progression of delivery of the three-dose series, it is clear from the map of DTP1 coverage in Fig. 1a that there is a moderate-to-high level of access to health and RI services in most places across the country, though some areas of very poor DTP1 coverage remain, suggesting low access to health and RI services or non-acceptance of vaccines in these areas. Also, there is a pronounced inability of the health system to follow-up with subsequent doses (impediments to re-attendance could also arise from other sources[17]), as reflected in poorer coverage levels for DTP3. Similar patterns exist in other countries, though modified by some distinctive features. In Nigeria, there is a major north–south divide with coverage levels for all doses remaining consistently low in most of the northern region and high in some parts of the south. In Ethiopia, there is a significant east–west divide (more pronounced in DTP1 coverage) occasioned by higher coverage levels in the west and lower levels in the east, much of which constitutes low population density areas. Also, coverage with DTP3 for this country seems particularly low in remote rural areas. In contrast, coverage levels in Cambodia and Mozambique were generally higher and more homogeneous, although some underperforming areas persist. These can be found in eastern Cambodia and central and northern parts of Mozambique and are more visible in the DTP3 coverage maps.

### Table 1 Model validation statistics

| Country | MCV | DTP1 | DTP2 | DTP3 |
|---|---|---|---|---|
| **Coverage (%)** | | | | |
| Nigeria | 80.43 | 84.39 | 82.23 | 82.66 |
| Cambodia | 90.31 | 95.71 | 96.53 | 92.41 |
| Mozambique | 88.67 | 87.39 | 88.24 | 86.55 |
| DRC | 72.15 | 63.14 | 67.21 | 67.22 |
| Ethiopia | 88.29 | 87.90 | 83.06 | 80.84 |
| **% Bias** | | | | |
| Nigeria | −1.47 | −1.70 | −2.07 | −2.23 |
| Cambodia | −0.81 | −0.14 | −0.41 | −0.83 |
| Mozambique | −0.50 | 0.11 | −0.14 | −0.30 |
| DRC | 0.54 | 0.37 | 0.39 | 0.44 |
| Ethiopia | −0.66 | −0.75 | −1.39 | -1.79 |
| **VMSE** | | | | |
| Nigeria | 0.02 | 0.03 | 0.04 | 0.05 |
| Cambodia | 0.02 | 0.01 | 0.02 | 0.03 |
| Mozambique | 0.02 | 0.01 | 0.02 | 0.04 |
| DRC | 0.04 | 0.04 | 0.05 | 0.10 |
| Ethiopia | 0.07 | 0.05 | 0.07 | 0.10 |
| **$R^2$** | | | | |
| Nigeria | 0.78 | 0.81 | 0.79 | 0.76 |
| Cambodia | 0.42 | 0.30 | 0.38 | 0.34 |
| Mozambique | 0.56 | 0.51 | 0.60 | 0.60 |
| DRC | 0.77 | 0.73 | 0.77 | 0.79 |
| Ethiopia | 0.50 | 0.59 | 0.61 | 0.60 |

Summary of model validation statistics based on a cross-validation exercise for all the countries. *VMSE* validation mean square error. The $R^2$ statistics are based on the fitted models in each case

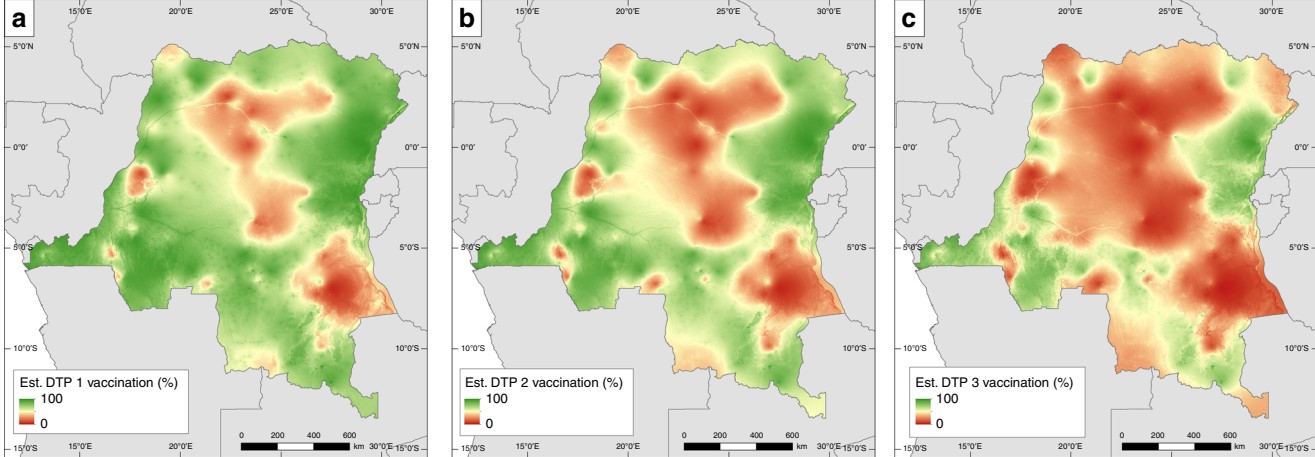

**Fig. 1** DTP1–3 vaccination coverage for DRC. Estimated DTP dose 1, 2 and 3 vaccination coverage in children under 5 years old at 1 × 1 km resolution for DRC in 2013–2014. Associated uncertainty maps are shown in Supplementary Fig. 1

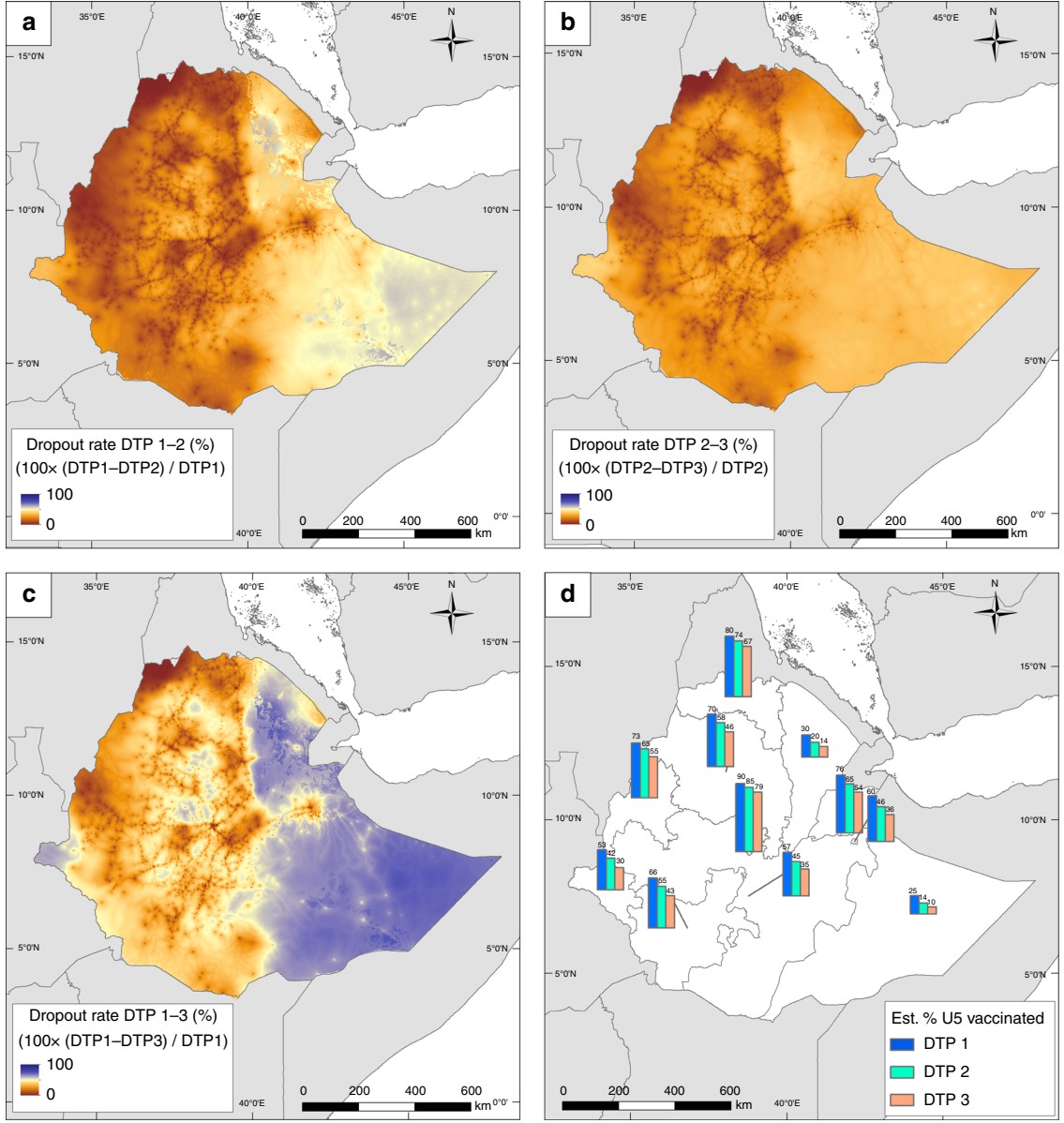

**Fig. 2** Dropout rates between DTP doses 1 to 3 in Ethiopia. **a–c** Estimated dropout rates between DTP vaccination doses 1 to 3 in children under 5 years old at 1×1 km resolution for Ethiopia in 2016. The estimated percentage of children receiving each dose in each administrative unit is shown in panel **d**

DTP dropout maps in Fig. 2 and Supplementary Figs. 7–10 (A–C) quantify the differences in the proportions of children who did not complete the three-dose series, dropping out either after the first or the second dose. As expected, the greatest geographical disparities in dropout rates occurred between DTP1 and DTP3 in all five countries, while the least disparities and lowest dropout rates were observed between doses one and two in DRC, Nigeria and Mozambique and doses two and three in Ethiopia and Cambodia. In DRC, Ethiopia and Nigeria, dropout rates between DTP1 and DTP3 were consistently highest in low coverage areas. Figure 2d shows the proportions of children aged under 5 years who received the respective doses in Ethiopia in 2016 at the provincial level (i.e. administrative level one); see panel D of Supplementary Figs. 7–10 for those of other countries. These regional statistics are population-weighted estimates obtained through integrating the coverage maps with grid-level United Nations-adjusted population counts obtained from WorldPop (www.worldpop.org)[20]. The same patterns as before can be seen in these regional estimates. In Ethiopia, for example, the

geographical divide in dose delivery is apparent among all three doses (Fig. 2d), with the percentage of children who received all three doses ranging between 30 and 79% in the western provinces and between 10 and 54% in the east. Interestingly, despite the overall higher coverage levels in the west, there remain some regions where considerable proportions of children (≥ 30%) have not received even the first dose (a similar pattern is also observed in eastern DRC). The capital, Addis Ababa, is the only region to have almost attained the 80% threshold with all three doses. The numbers of zero-dose and under-vaccinated (i.e. those who started the first dose but did not complete the series) children varied according to the sizes and population densities of the areas (see Supplementary Table 9). In Ethiopia, these ranged from 14,255 and 8,875 children in Harari to as high as 2,267,627 and 1,196,071 children in Oromia, respectively.

At 1×1 km grid level, Fig. 3 highlights areas within each country where substantial challenges exist in meeting the 80% coverage threshold as set out by the Global Vaccine Action Plan (GVAP) with all three DTP doses. In Cambodia, much of the

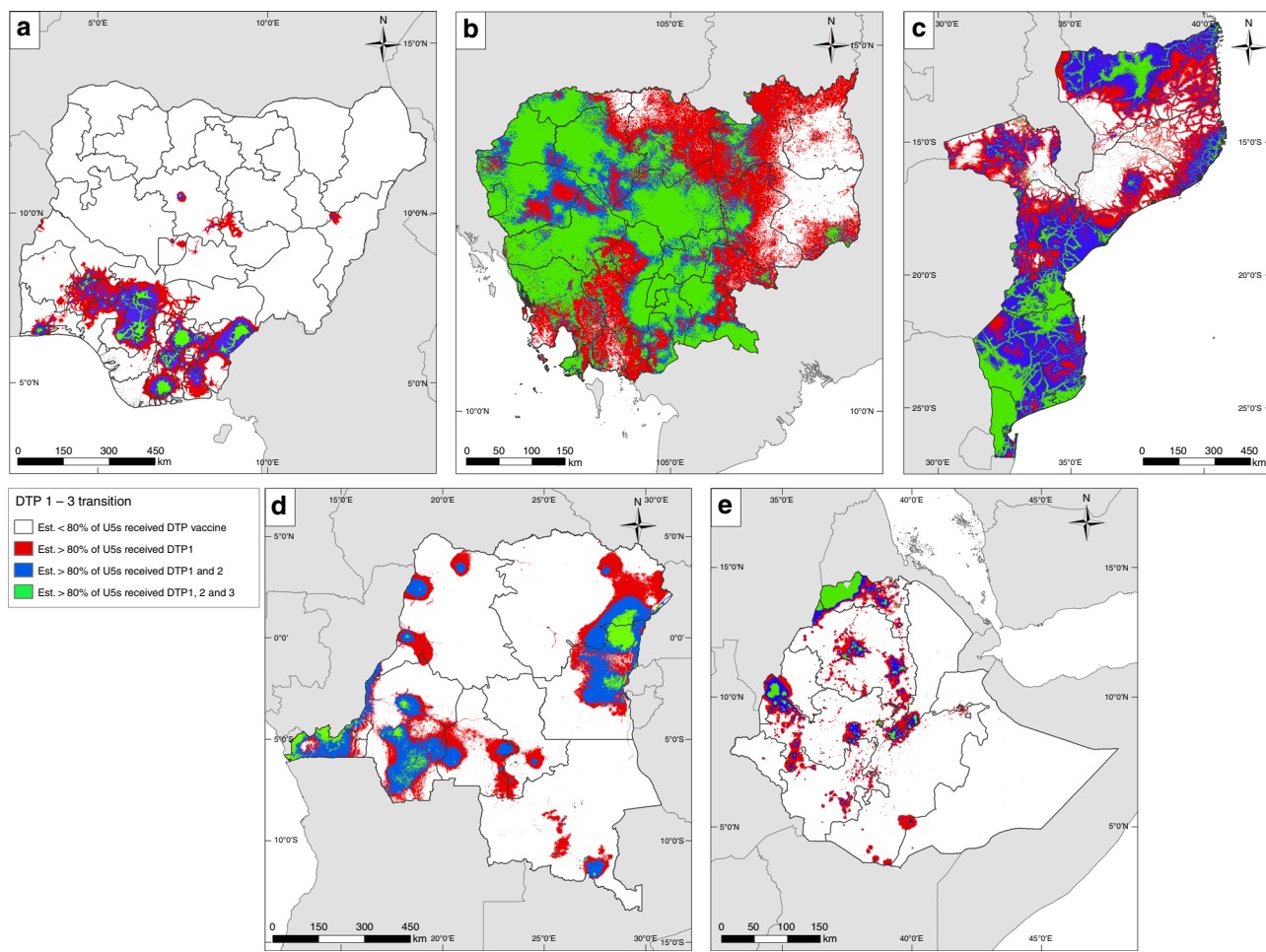

**Fig. 3** Attainment of 80% coverage with DTP doses 1 to 3. Transitions in coverage from DTP doses 1 to 3 in **a** Nigeria (2013), **b** Cambodia (2014), **c** Mozambique (2011), **d** DRC (2013–2014) and **e** Ethiopia (2016), highlighting areas where over 80% of children aged under 5 years have been vaccinated in accordance with the WHO Global Vaccine Action Plan targets. Areas in green are those where >80% of children are predicted to have received all three doses of DTP vaccine

country is compliant except the eastern area where the delivery of the last two doses is lower than expected. There is a north–south gradient in Mozambique with the southern part of the country being successful in reaching the threshold with the second dose (but the third dose appears to fall short of the target in some areas) whereas the north constitutes a problematic area, particularly in delivering the last two doses. Vast extents of Nigeria, DRC and Ethiopia have not attained the target with any of the doses, but there are pockets of areas where demonstrable successes have been achieved. These include Addis Ababa and parts of Tigray and eastern regions of Ethiopia, parts of southern Nigeria and areas in eastern and western DRC.

**Comparing MCV with DTP3 coverage maps.** Interesting patterns are apparent in MCV coverage in these countries (Fig. 4 and Supplementary Fig. 6). While coverage levels are generally high and more spatially homogeneous in Cambodia and Mozambique, significant heterogeneities exist in Nigeria, Ethiopia and DRC. There are areas of similarities in the coverage of MCV and DTP3 vaccinations (e.g. high MCV, high DTP3 and low MCV, low DTP3 areas) across all five countries, but striking differences are also evident (Fig. 4, left and middle panels), with MCV having higher coverage levels in more areas (as is also depicted in Supplementary Figs. 12 and 13). More specifically, in DRC, coverage with MCV is substantially higher than with DTP3, with areas of

higher MCV coverage constituting about 99% of the geographical extent of the country (see right panel of Fig. 4) and having an average percentage difference of 55.21% (the percentage differences were calculated relative to the average coverage of the two vaccines at 50 km$^2$ resolution). In Ethiopia, there are marked similarities in coverage gradient between the two vaccinations, with MCV showing higher coverage over 87% of the country with an average percentage difference of 58.67%, although there are pockets of areas where DTP3 performed better. A similar pattern is evident in Nigeria: differential in coverage between northern and southern regions is similar between the two vaccines, and MCV has an overall higher coverage across 89% of the country where an average percentage difference of 40.63% was obtained, although some areas with higher DTP3 coverage exist. In Cambodia, both vaccinations have distinctive high coverage levels with clusters of areas of slightly higher MCV (65% of the country with an average percentage difference of 7.57%) and slightly higher DTP3 spreading across the country. For Mozambique, areas of slightly higher MCV coverage forming about 68% of the country with an average percentage difference of 10.9% are concentrated in the northern region, whereas areas of slightly higher DTP3 coverage are mostly located in the middle and southern regions. In general, although there are consistent areas of low and high coverage for both vaccines, significantly higher coverage with MCV was observed in DRC, Ethiopia and Nigeria compared to Cambodia and Mozambique.

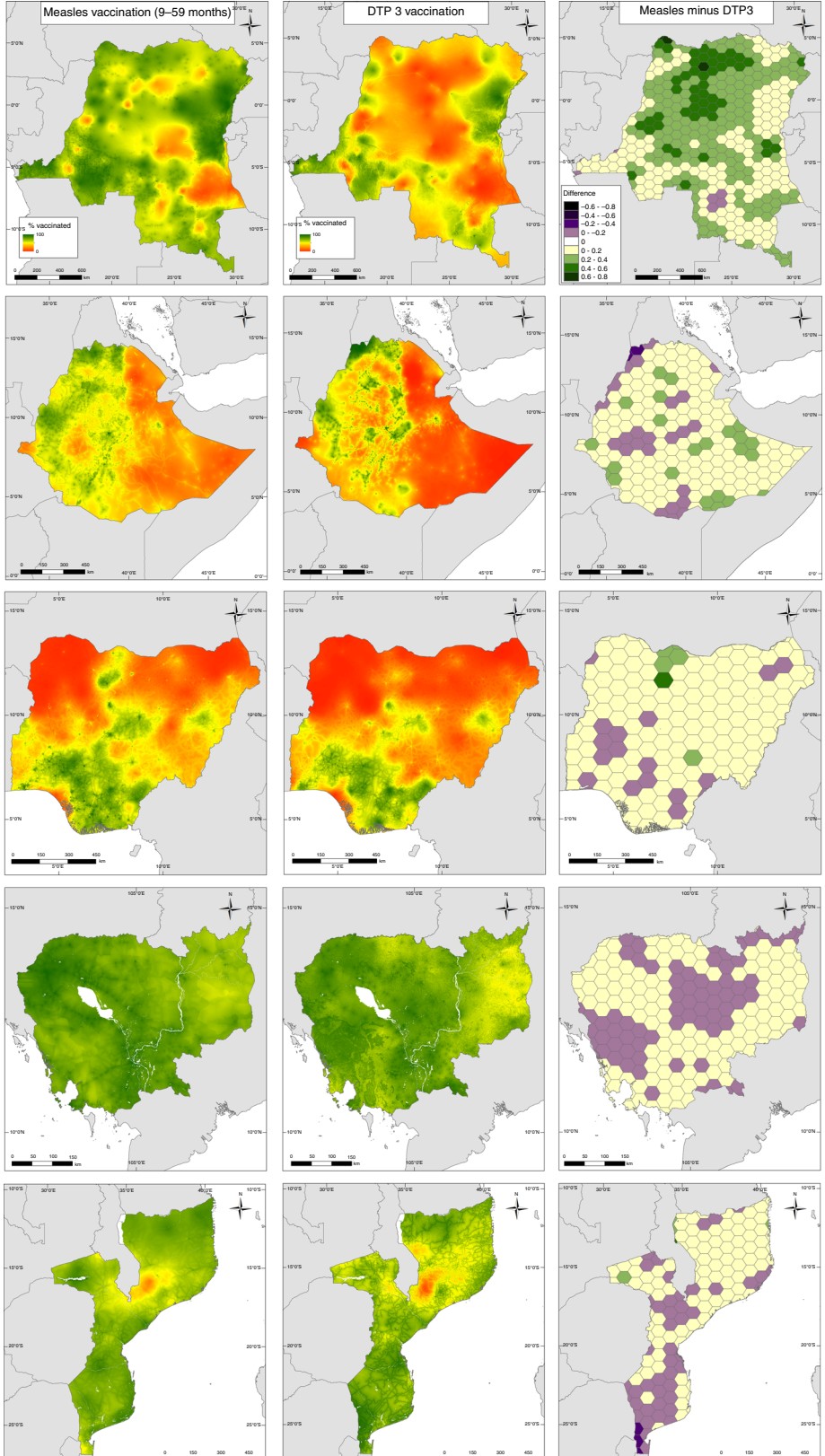

**Fig. 4** MCV versus DTP3 coverage. Comparison between predicted measles (left panel) and DTP3 (middle panel) vaccination coverage in children aged 9–59 months and under 5 years old, respectively, at 1 × 1 km resolution in (top-bottom) DRC (2013–2014), Ethiopia (2016), Nigeria (2013), Cambodia (2014) and Mozambique (2011). Right panel: Differences between predicted measles and DTP3 coverage, with results aggregated to a 50 km² hexagon grid for ease of visualization. The maps of the differences at 1 × 1 km are shown as Supplementary Fig. 13

## Discussion

The value of spatially detailed data is increasingly being recognized by the international health and development community. The Sustainable Development Goals were launched in 2015 with the central theme of 'leaving no one behind'[21]. In addition, the GVAP[22] has set a target of reaching 80% coverage with all vaccines in every district by 2020. With nonattainment of vaccination targets in many areas[1,2] and the need to improve access to vaccinations for marginalized populations[22], this work provides a method to guide the optimal combination of vaccine delivery strategies to accelerate progress towards coverage targets. The quantification of disparities in coverage levels and the successes and failures of vaccine delivery strategies at spatially resolved levels offers the flexibility for programmatic utilization of outputs at more operationally relevant spatial scales within countries.

The $1 \times 1$ km predicted maps showed substantial gaps in the coverage of DTP1 particularly in Nigeria, DRC and Ethiopia—suggesting poor access to or utilization of RI services (including vaccine stockouts in existing health facilities), vaccine refusal or lack of awareness about vaccines, low parental education and non-vaccination due to mothers' social engagements[23–25], coupled with a clear evidence of lack of health system continuity in these countries. The estimated geographical disparities in DTP dropouts and their varying degrees of occurrence reflect potential differences in the determinants of incomplete vaccination[24,26–30] within and across these countries. Problems underlying non-receipt of the three-dose series in areas with high DTP1 coverage (e.g., type of health facility attended[31], missed opportunities[32], bad experiences at immunization centres[32], poor maternal/caregiver knowledge of immunization schedules[33] and inactive defaulter tracing mechanisms[34]) are likely to be different from factors responsible for poorer coverage with the latter doses in areas with low DTP1 coverage (e.g., difficult terrains, unavailability of vaccine, interruptions due to conflicts[35], poor parental education[30] and inadequate health infrastructure[36]). The drivers of higher dropouts between the first two doses in some areas are also possibly different from the drivers of higher dropouts between the last two doses in others. In addition, a previous study[15] found that whereas maternal education was a determinant of both dropouts between DTP1 and 3 and non-vaccination with any dose, access to health facilities was only associated with the latter. Hence, there is a need for contextual, in-depth assessments of barriers to complete vaccination with DTP doses in these countries in light of the estimated coverage patterns. Also, the confluence of high population density and poor DTP1 and/or DTP3 coverage in some areas shows a high density of un- or under-vaccinated children who are therefore at risk of outbreaks. This underscores the importance of health and immunization system strengthening in the identified areas in these countries[37].

Substantially higher MCV coverage compared to DTP3 coverage in much of DRC and parts of Ethiopia and Nigeria provides evidence that the supplementation of RI with various SIA campaigns (see Supplementary Tables 8 and 13) has been effective in these countries. Specifically, DRC had 25 campaigns during the reference period of the DHS survey analysed, while Ethiopia and Nigeria each had five campaigns, all of which included both national and subnational campaigns. For Cambodia and Mozambique, with only three and two national campaigns respectively, no marked improvement in MCV coverage over DTP3 was seen at the national level. Therefore, unlike DRC, Ethiopia and Nigeria where RI systems appeared weaker, measles SIA campaigns had not greatly improved coverage over the RI system in Cambodia and Mozambique (though the SIAs likely increased immunity among children for whom the SIA dose may have been a second dose—this level of detail is not captured in

DHS surveys). At the local level, the occurrence of low MCV coverage in areas with low DTP3 coverage across these countries generally indicates a lack of the impact of SIAs in these areas, thus highlighting the need for intensified efforts in these areas. Furthermore, the broader success of vaccine delivery strategies lies in achieving coverage levels high enough to interrupt disease transmission[38]. Our assessments at both fine spatial scales and regional levels (Figs. 2d and 3, Supplementary Figs. 7D–10D) using all three DTP doses as example cases showed that large areas of Nigeria, DRC and Ethiopia and parts of eastern Cambodia and northern Mozambique had not attained the threshold of 80% with any of the doses (and much higher coverage is needed for measles elimination). Thus, the potential for continued disease circulation and outbreaks remains high in these areas.

Our analysis is subject to some limitations. The accuracy of the maps produced is largely dependent upon the accuracy of the data used. There is a potential for information bias in the vaccination coverage data arising from determining vaccination status using parental recall in the absence of home-based records[39], and the rather broad age groups used to evaluate the coverage of both MCV and DTP vaccinations. Including all children aged under 5 years in the DTP analysis will slightly underestimate coverage since those under 14 weeks of age would not be eligible for the third dose of DTP (<5% and <11% of surveyed children were aged ≤1 month and ≤3 months, respectively, in each of the study countries, see Supplementary Table 10). Intuitively, mothers are less likely to remember the number of vaccine doses their older children have received, especially in the case of DTP doses which are administered during early childhood. Analysis based on data obtained from vaccination cards only would be hampered by the large proportions of children without vaccination cards in the study countries and sample size issues—small sample sizes at the cluster level diminish the predictive power of the fitted models[2]. There is also the possibility of an upward bias in coverage levels attributable to the representativeness of DHS surveys as these rely on samples drawn using census maps which may be outdated, incomplete or miss important underserved areas.

The output maps produced here relate to the dates of the respective DHS surveys. This implies that the effects of more recent vaccination activities are not captured in the analysis. More up-to-date maps can be obtained either through more recent survey data or other modelling and forecasting techniques. Apart from SIAs, unmeasured attitudinal factors not included in this study, such as mothers/caregivers being more concerned about measles and hence more likely to have their children/wards vaccinated against it compared to other diseases, could also give rise to a comparatively higher MCV coverage. On the other hand, better DTP3 performance could be due to the fact that it is easier for the health system to follow-up with the administration of DTP doses in the first 4 months of life compared to MCV which is given much later (from age 9 months) in the sequence of recommended vaccines. Geographical inequities in vaccination coverage could be dependent on many other factors, which include ethnicity, vaccine refusal, stockouts and maternal educational status[6,11]. The inclusion of these covariates in the modelling process, where their gridded surfaces are available, could enhance the predictive ability of the models and the exploration of the effects of delivery mechanisms undertaken here. Alternatively, methodological approaches enabling the joint prediction of these covariates and vaccination coverage, where relevant data exist at the survey cluster locations, could also be explored.

The methodology used in this work has utilized open data and software, demonstrating the potential for application to other

LMICs and other childhood vaccines to enable more antigen- or vaccine-specific assessments. We estimated vaccination coverage in children aged under five years; however, an age-structured approach (see ref. [2]) to evaluating the impact of delivery mechanisms will highlight areas where vulnerable subpopulations exist within countries and provide more precise information to strengthen program planning. Although limited information did not permit the exploration of this approach in the current analysis (e.g., it is difficult to determine which children participated in subnational SIAs using DHS data), we will seek multiple sources of data to undertake this in future work. Additionally, integrating key variables such as treatment-seeking behaviour[40], travel time to health facilities and mobile network data[41] into future analysis could improve the identification of mobile and other hard-to-reach populations to facilitate the design and implementation of tailored vaccine delivery programs. The predictive performance of the fitted models has been evaluated using cross-validation methods. However, the predicted maps could be further validated by using external data sources such as the WHO EPI surveys[42] or Multiple Indicator Cluster Surveys (mics.unicef. org). Also, an integrated study investigating infectious disease dynamics and vaccination coverage could reveal further insights on the performance of vaccination programs in the study countries.

In conclusion, the work undertaken here has highlighted significant geographical variations in the effectiveness of vaccine delivery mechanisms in the study countries. We have shown areas of low coverage due to inadequate RI systems manifesting as poor access to vaccination services and inability to follow-up with latter vaccine doses. Importantly, we have delineated areas in these countries where SIA campaigns have been effective in boosting coverage levels substantially beyond that achieved through RI, especially in DRC where recurrent SIAs occurred, and parts of Nigeria and Ethiopia. Intensified and more targeted strategies are therefore required to improve coverage levels in all underperforming areas, particularly where there is a coincidence of poor RI and SIA performance if disease elimination goals must be reached. In terms of achieving coverage targets, this study has shown, using all three doses of DTP as an indicator, that much of DRC, Ethiopia and Nigeria were yet to achieve the GVAP target of 80% coverage at the time of the respective surveys, whereas Cambodia and Mozambique had attained the target with some doses in many areas. Thus, while the output maps and findings on delivery strategies are useful for developing vaccination strategies in pursuit of national vaccination targets in the first three countries, more localized interventions based on these results in the last two countries will help improve regional coverage levels and accelerate progress towards disease elimination.

## Methods

**Vaccination coverage and covariate data.** Data on the coverage of MCV and DTP1–3 vaccinations in children aged under 5 years in these countries were obtained from the Demographic and Health Surveys (DHS) database[43]. The most recent datasets available were used for the analysis—surveys were conducted in 2011 (Mozambique), 2013 (Nigeria), 2014 (Cambodia and DRC) and 2016 (Ethiopia). These countries follow WHO recommendations hence DTP is recommended from ages 6, 10 and 14 weeks and MCV from age 9 months. For MCV, we focus on coverage with at least the first dose as this is the definition of MCV coverage used by the DHS program[43–45] and only Cambodia had included a routine second dose of MCV in its program by the time of the survey; whereas for DTP, coverage is determined for the individual doses. For each surveyed child aged under 5 years, the vaccination status, age in months at the time of the survey, and the global positioning system (GPS) coordinates (longitude and latitude) of the survey cluster to which the child's household belonged were extracted. The DHS ascertains vaccination status for each vaccine-dose combination from home-based records (or vaccination cards) when available, from parental recall, and in the case of Ethiopia, from health facility records. We used the data for coverage of each vaccine dose by the time of the survey including all sources of information (card, recall, health facility). For each vaccine dose, the data extracted were summarized at

the survey cluster level, which is the geographic unit of our analysis—see Supplementary Fig. 11 for the locations of these clusters and the corresponding observed coverage rates. It is pertinent to highlight here that although DHS surveys are designed to be representative at much coarser geographies, the use of geolocated cluster-level data in the production of high-resolution maps of indicators is now a well-established practice[1,2,46,47]. The GPS coordinates of DHS clusters are usually displaced up to 2 km in urban areas and up to 5 km in rural areas to protect the confidentiality of respondents. We accounted for this displacement during covariate extraction by using buffers to ensure that the correct cluster centroid was included in the analysis. Following recommended approaches[48], we created a 5 km buffer around clusters in rural areas and a 2 km buffer around those in urban areas.

To contextualize our comparison of MCV and DTP3 coverage, information on relevant measles SIAs occurring in the 5 years preceding the surveys, according to WHO records[13] are reported in Supplementary Table 8. Whereas DRC had about 25 campaigns—2 national and 23 subnational campaigns, other countries had at least 2 national campaigns and a total of between 2 and 5 campaigns during the respective periods of interest. In addition, these campaigns targeted different age cohorts of children and their nature varied depending on whether they were prophylactic or outbreak response strategies. In order to ensure that we capture as much as possible all surveyed children who may have participated in these SIAs, our analysis of MCV coverage included all children aged 9–59 months at the time of each survey. However, for DTP, our analysis is based on all available data which relate to all children aged under 5 years.

Following previous work[2], to inform our prediction models, several geospatial socio-economic, environmental, and physical factors known to be directly or indirectly associated with the spatial distribution of vaccination coverage were assembled. These included metrics related to remoteness, poverty, livestock, slope, and land cover; see Utazi et al.[2] and Supplementary Tables 1 and 2. From these datasets, a series of standardised gridded covariate layers were constructed at $1 \times 1$ km resolution for all countries, using the most recent versions of datasets available and where possible, those closest to the year of each country's DHS survey using ArcGIS v10.4. At each DHS cluster location, covariate data values were extracted from the each of the standardised layers. For continuous covariates, the extracted data were the means of the values falling within the buffers. Where a covariate was categorical in nature, the majority class of cells falling within the buffer was extracted. If a covariate layer was binary, the maximum value was extracted to determine presence/absence of the covariate within each buffer.

Covariate selection was performed using approaches detailed in Utazi et al.[2] to determine the best combination of covariates for modelling MCV and DTP1–3 vaccination coverage. This involved examining the relationships between each covariate and vaccination coverage and applying the log transformation to the covariates where necessary, testing for multicollinearity and fitting non-spatial univariate binomial generalized linear models in a frequentist framework. The non-spatial models used here is a standard practice based on the notion of explaining as much of the variation in the data as possible using available covariates before accounting for any residual spatial autocorrelation. The selected covariates were determined using the $p$-value approach which is relative to backward elimination[49]. For each country, we formed a uniform set of covariates for DTP1–3 from the initial set of covariates selected for each dose – this step was not required for MCV.

**Mapping MCV and DTP1–3 coverage.** The modelling framework used in this work builds on a previously developed framework[2]. Let $s_i$ ($i=1, \ldots, n$) denote the cluster locations included in a given survey, $N(s_i)$ the corresponding number of children sampled and $Y(s_i)$ the number who have received a given vaccination. Our aim is to predict $p(s_i)$, the probability of being vaccinated at location $s_i$, over $1 \times 1$ km grid locations covering each country. We model $p(s_i)$ corresponding to the cluster locations using a binomial spatial regression model with a logit link, and a linear combination of covariates and a set of spatial random effects $w=(w(s_1),\ldots,w(s_n))^T$ modelling autocorrelation in the data included in the linear predictor. For measles vaccination, each country's data were modelled using the univariate approach as in Utazi et al.[2], with $w$ specified as a zero-mean stationary Gaussian process with an exponential covariance function.

For DTP vaccination, the univariate model leads to challenges when calculating the dropout rates between the doses due to the occurrence of predicted coverage levels for the latter doses higher than for the earlier doses in many areas. Hence, we adopted a multivariate version of the model to jointly model the three doses, which yielded improved results. Using the notation described previously, the multivariate model can be expressed as

$$Y_j(s_i)|N_j(s_i) \sim \text{Binomial}\,(N_j(s_i), p_j(s_i)), \quad j = 1, 2, 3,$$
$$\text{logit}(p_j(s_i)) = \mathbf{x}_j(s_i)^T \boldsymbol{\beta}_j + w_j(s_i), \tag{1}$$

where $j$ indexes the doses and $\mathbf{x}_j(.)$ and $\boldsymbol{\beta}_j \in R^k$ are the respective covariate vector and regression parameters for the $j$th dose. For a generic location $s$, the process $\mathbf{w}(s)=(w_1(s), w_2(s), w_3(s))^T$ is a zero-mean multivariate Gaussian process with cross-covariance function $\boldsymbol{C}_w$ used to account for spatial variation in the model. The elements of $\boldsymbol{C}_w$ are such that for any pair of locations $s$ and $s'$, $\boldsymbol{C}_w(s, s') = \text{cov}(w_j(s), w_{j'}(s'))$, $(j, j' = 1, 2, 3)$. For $n$ cluster locations, the $3n \times 1$ vector $\boldsymbol{W} = (\mathbf{w}^T(s_1), \ldots, \mathbf{w}^T(s_n))$ follows a multivariate normal distribution $\boldsymbol{W} \sim MVN(0, \boldsymbol{\Sigma}_w)$, where $\boldsymbol{\Sigma}_w = [\boldsymbol{C}_w(s_i, s_{i'})]_{i,i'=1}^n$ is a $3n \times 3n$ matrix that can be

partitioned as an $n \times n$ block matrix, with the $ij$th block being the $3 \times 3$ cross-covariance matrix $C_w(s_i, s_j)$. The cross-covariance function $C_w$ was specified using the coregionalization approach (see refs. [50,51]), which includes for each dose, an exponential correlation function with decay parameter $\phi_j$ (see ref. [2]), determining the effective ranges (i.e. the distance at which spatial correlation reduces to 0.05, calculated as $-\log(0.05)/\phi_j$) of the process $w(s)$ through an underlying spatial process. Further details of the model including prior specification are provided in Supplementary Methods. The Bayesian model in (1) was implemented via Markov Chain Monte Carlo (MCMC) methods using spBayes package in R[52,53]. For ease of computation, particularly in the case of the multivariate model for DTP1–3, we used the predictive process model specification in the package throughout this work; see ref. [52] and Supplementary Methods for details.

For each country, the fitted models were used to predict $p(s)$ for both MCV and DTP1–3 over $1 \times 1$ km grids. Predictions for more operationally relevant administrative areas were obtained by averaging $p(s)$ over the $1 \times 1$ km grids within each area. Following a Monte Carlo cross-validation exercise repeated 10 times with a 10% subset of the data used for validation each time, the percentage bias, VMSE and the coverage percentage of the 95% credible intervals of the predictions (i.e., the proportion of times (in percentage) that the 95% credible intervals of the predictions contained the true values) from the models for both MCV and DTP1–3 were used to assess their performance for out-of-sample prediction (see Supplementary Methods for details). While percentage bias and VMSE measure the accuracy of the central tendencies of the predictions, the coverage percentage measures the correctness of their uncertainties. Also, the coefficients of determination ($R^2$) of the fitted models were used to evaluate their predictive ability in each case.

**Reporting Summary**. Further information on experimental design is available in the Nature Research Reporting Summary linked to this article.

## Data availability

All data used in this work is publicly available via the sources referenced in the Methods section. These can be obtained from the authors upon request.

## Code availability

The R code used in the analysis is available at: https://github.com/EdsonUtazi/NCOMMS_paper_code.

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

## Acknowledgements

This work is supported by the Bill & Melinda Gates Foundation (to C.J.E.M., M.J.F., A.J.T. and J.L.) through grant number: OPP1094793. A.J.T. is supported by funding from NIH/NIAID (U19AI089674), the Bill & Melinda Gates Foundation (OPP1106427, 1032350, OPP1134076), the Clinton Health Access Initiative, National Institutes of Health and a Wellcome Trust Sustaining Health Grant (106866/Z/15/Z).

## Author contributions

C.E.U., J.T., V.A.A., M.J.F., S.T., C.J.E.M., J.L., F.T.C. and A.J.T. conceived and designed the study. C.E.U. and J.T. performed the data analysis. C.E.U. and A.J.T. wrote the first draft of the manuscript. C.E.U, J.T., V.A.A., M.J.F., S.T., C.J.E.M., J.L., F.T.C. and A.J.T. contributed to the writing and editing of the manuscript.
