## [Peer Review File · Nature Communications]

Reviewers' comments:

Reviewer #2 (Remarks to the Author):

SUMMARY

This manuscript describes a study that used data from Demographic and Health Surveys about measles (MCV) and DTP vaccination coverage and additional data to create 1x1 km raster maps of vaccination coverage for five countries (Nigeria, Ethiopia, DRC, Cambodia, Mozambique). Because supplementary immunization campaigns (SIA's) are conducted in some countries for measles but not for diphtheria, the study aimed to explore the effect of SIA's in addition to routine immunization (RI) by comparing MCV vs. DTP3 coverage rates. The methods for using DHS data to make high-resolution maps have been used previously and the study seems overall sound. Using MCV compared to DTP3 to make inference about SIA's is a bit problematic since many areas in study countries did not have SIA campaigns, so the 1x1 km maps of MCV vs. DTP3 could not be systematically used to assess the effect of SIA's beyond some coarse descriptive statements. The high resolution maps could therefore mostly be used to describe subnational vaccination coverage patterns rather than to "explore delivery mechanisms" as mentioned in the title. The conclusions about the delivery mechanisms, i.e., that areas with poor coverage did not have effective SIA's, was not surprising either. The high-resolution vaccination coverage maps are valuable to assess areas of risk for outbreaks and to inform vaccination strategies. I recommend publication of this study with the focus on the vaccination coverage maps (and not on comparing delivery mechanisms) after some changes have been made.

MAJOR COMMENTS

- The high-resolution maps of vaccination coverage are useful, but it would be helpful to summarize any generalizable findings from these maps into conclusions about the spatial patterns in vaccination coverage that could help operational decision making.
- In addition to mapping MCV and DTP, the study explores dropout rates in DTP vaccination, between the various doses. The added value of the dropout analysis could be better described in the introduction and discussion sections.
- The objective of this study, as reflected by the title and described in the introduction, is to compare vaccination delivery mechanisms, i.e., RI and SIA's by comparing MCV and DTP3 coverage. The only results presented for this comparison is a coarse description comparing MCV and DTP3 coverage maps. If I understand correctly, the only data on SIA's included in this study is on their frequency and the % coverage as presented in Table S7. Given that many districts in the study countries did not implement SIA's, it would not be possible to conduct a systematic, statistical analysis comparing 1x1 km maps of MCV and DTP3 coverage in areas with and without SIA's to assess the effect of SIA as delivery mechanism. If the only analysis possible, given the data, of delivery mechanisms, is the current description comparing MCV and DTP3 coverage maps, I would recommend to not make this comparison the primary focus of the paper. If a more formal, statistical analysis is possible with available data, then I would recommend presenting that analysis instead of the current description, to strengthen the paper with its current focus.
- The authors discuss the difference in coverage between DTP3 and MCV, but did not quantify this in the results section. Would it be possible to show maps/statistics of the difference between DTP3 and MCV at the 1x1 km resolution and then identify areas where RI is clearly underperforming (stating the limitation that not all areas conducted SIA's)?
- There was a substantial difference in model fit for DTP vs. MCV with much better fits for MCV, at least from R2 values. Please explain this difference in model fit and what possible, unmodeled, factors could play a role in DTP but not in MCV?
- As far as I understand, DHS data are sampled to represent statistics at the regional level, which

is a much coarser spatial resolution than the 1x1 km grid estimates used in this study. Why is it appropriate to make inference at the 1x1 km resolution when DHS data were sampled to represent a much higher resolution? And do the DHS data allow specific inference of spatial statistics at a higher resolution, e.g., as made in line 211:212 "The capital.... all three doses"?

- The uncertainty maps for some countries, such as DRC, show a very distinct dotted pattern, with high uncertainty in some very specific places. What is the explanation for this pattern? And, assuming that the uncertainty levels are mostly related to the location of the DHS clusters (more uncertainty in areas without DHS clusters), could the authors also show a map of each country with the location of the clusters?

- Table S1: It would help the interpretation of the models if the covariates could be described specifically, e.g., what exact poverty indicator, and the units of each covariate. Also, it is unclear why different variables were included for each country; what was the variable selection process?

MINOR COMMENTS

- Introduction lines 57:59 "Understanding the spatial dynamics....programs": it will indeed be valuable to better understand spatial dynamics in complex environments with infectious disease transmission and various intervention programs, but this study did not evaluate dynamic, but rather static images of vaccination coverage. This study also separately analyzed each indicator (DTP3 and MCV), instead of a analysis of dynamic patterns of disease and both vaccination strategies (e.g., representing changes over time and interactions between vaccination strategies). It may be beyond the scope of this manuscript to study dynamic patterns, but given the statement on their importance in the introduction, the paper should probably mention the design/components of a study that would represent dynamic processes.

- Line 105 "strong association": consider replacement with "statistically significant association".

- Lines 119-134 "The parameters...as expected": consider moving this section to the discussion, it distracts from the story and the stronger spatial dependence for DTP1 vs. 3 is explained as with the dropout process, which is not the main focus of the manuscript, but a secondary focus.

- Lines 137:141 "Table 1...Ethiopia": The bias should be summarized separately for MCV and DTP, since these are the programs under comparison.

- Line 141 "VMSE": VMSE should be written in full at first mention in the text.

- Lines 143-148 "For Cambodia.... 94.9%": This sentence should be written more clearly and precisely, e.g., it is not clear what "achieved coverages of 95% prediction intervals were close to the true value of 95%" means. The following sentences inherit some of this lack of clarity. Maybe it would help to first introduce the concept of comparing 95% prediction intervals of model estimates with 95% confidence intervals of the observed data, or something similar, before listing the values for each country.

- Lines 307-309 "There are also biases.... underserved areas": It would be helpful to describe these biases in a bit more detail, particularly the direction in which these biases could have affected study results.

- Lines 314-317 "Geographical.... educational status": The authors included various covariates in their vaccine coverage prediction models, such as remoteness or environmental factors, so these associations could shed light on what factors drive vaccination coverage. If other factors may exist that are statistically significantly associated with vaccination coverage, these would improve the spatial prediction models as well. In that context, the authors should explain why they selected the covariables currently included in the model, but left others for future work. I understand that no study can measure the effect of all possible covariates, but would it not make sense to explore

major factors such as education, ethnicity, and e.g., nutritional status, for possible inclusion in prediction models of vaccination coverage? In that context, it would be helpful to describe the variable selection process in more detail in the methods section (see also previous comment about that).

- Lines 269-273 "Problems...infrastructure": Given that DTP doses are spaced very closely (within months of each other), it is unlikely that long-term processes such as conflicts, may be the main determinant of missing doses. Would it not be expected that the same factors explain low coverage and missing doses, such as remoteness, poverty, low education, etc.? A specific example contrasting determinants of completing a schedule vs. overall coverage, or additional citations, would strengthen this section.

- Supplement line 43 "validation locations": Just to be sure, are these locations the DHS clusters or the 1x1 km grid cells? From the description, I would assume these would be the 1x1 km grid cells.

Reviewer #3 (Remarks to the Author):

More accurate and finer resolution maps of vaccination coverage can help guide improvements in the provision of routine and supplementary immunisation services. This paper describes the use of a Bayesian statistical model to predict vaccination coverage in 5 countries at a 1km resolution using high-resolution spatial covariates and vaccination coverage information from demographic health survey (DHS) data. The performance of these methods is assessed using cross-validation.

This is important work and I am supportive of publication. I have only a few comments/questions:

The reliance on DHS data should be made clear in the abstract and introduction, and perhaps also the caveat (already mentioned in the Discussion) that extrapolating to areas without DHS cluster data can be problematic when there are areas which cannot be surveyed (e.g. parts of northern Nigeria including Borno state).

It is not clear to me why a non-spatial model is used to perform model selection in terms of included covariates. Is this simply to save computation time?

There are opportunities to further validate these coverage maps aside from using cross-validation methods. This includes the use of alternative immunisation coverage data (e.g. WHO surveys, polio surveillance data). There are also opportunities to compare coverage with incidence of disease (e.g. measles). Have the authors plans to include such data? Could this be mentioned?

Response to reviewers' comments

Response to comments by Reviewer #2

General comment: Using MCV compared to DTP3 to make inference about SIA's is a bit problematic since many areas in study countries did not have SIA campaigns, so the 1x1 km maps of MCV vs. DTP3 could not be systematically used to assess the effect of SIA's beyond some coarse descriptive statements. The high resolution maps could therefore mostly be used to describe subnational vaccination coverage patterns rather than to "explore delivery mechanisms" as mentioned in the title. The conclusions about the delivery mechanisms, i.e., that areas with poor coverage did not have effective SIA's, was not surprising either. The high-resolution vaccination coverage maps are valuable to assess areas of risk for outbreaks and to inform vaccination strategies. I recommend publication of this study with the focus on the vaccination coverage maps (and not on comparing delivery mechanisms) after some changes have been made.

Response: Thank you for these comments. According to WHO records reported in Supplementary Table 7, all five study countries had at least two **national SIAs** within the reference periods of the respective DHS surveys analysed in our work. Additionally, DRC, Ethiopia and Nigeria had some **subnational campaigns** within these periods.

Since these SIAs occurred within the reference periods of the surveys, improvements in coverage due to these activities, if any, would have been captured by the surveys, and propagated in our analysis to the predicted maps through the modelled data. This is what the comparison of MCV and DTP coverage undertaken in our work is based upon.

However, we have added the use of our analyses to inform vaccination strategies in the title. Also, as we have highlighted in the Discussion section, apart from SIAs, unmeasured attitudinal factors not included in this study, such as mothers/caregivers being more concerned about measles and hence, more likely to have their children/wards vaccinated against it compared to other diseases, may also give rise to comparatively higher MCV coverage. Similarly, better DTP3 performance could be due to the fact that it is easier for the health system to follow up with the administration of DTP doses after birth compared to MCV which is given much later (from age 9 months) in the sequence of recommended vaccines.

Following Major Comment #3, more statistics are now reported in the Results section (Comparing DTP3 with MCV coverage maps) to enhance the quantification of the differences between MCV and DTP3 coverage.

Major comments

Comment #1 The high-resolution maps of vaccination coverage are useful, but it would be helpful to summarize any generalizable findings from these maps into conclusions about the spatial patterns in vaccination coverage that could help operational decision making.

Response: Thank you for this comment. We have endeavoured to summarize generalizable findings, and the spatial patterns in the predicted maps are explained in detail in the Results section – both the patterns in the comparisons between MCV and DTP3 and in DTP1-3 coverage. General patterns from the maps are summarized in the second, third and last paragraphs of the Discussion section.

Comment #2 In addition to mapping MCV and DTP, the study explores dropout rates in DTP vaccination, between the various doses. The added value of the dropout analysis could be better described in the introduction and discussion sections.

Response: Thank you for this comment. We did mention the utility of the dropout rate analysis in the introduction to include the assessment of health system continuity. We have now highlighted that we produced estimates of numbers of under-vaccinated children (i.e. children who received the first dose but not the second dose, etc) and assessed progress towards coverage targets using the output maps.

In the discussion, we have additionally noted, following the results reported in Supplementary Table 8, that the confluence of high population density and low DTP1 coverage and/or high dropout rates between the doses implies that many children remain unvaccinated or under-vaccinated, which highlights the importance of strengthening of health systems in these countries.

Comment #3 The objective of this study, as reflected by the title and described in the introduction, is to compare vaccination delivery mechanisms, i.e., RI and SIA's by comparing MCV and DTP3 coverage. The only results presented for this comparison is a coarse description comparing MCV and DTP3 coverage maps. If I understand correctly, the only data on SIA's included in this study is on their frequency and the % coverage as presented in Table S7. Given that many districts in the study countries did not implement SIA's, it would not be possible to conduct a systematic, statistical analysis comparing 1x1 km maps of MCV and DTP3 coverage in areas with and without SIA's to assess the effect of SIA as delivery mechanism. If the only analysis possible, given the data, of delivery mechanisms, is the current description comparing MCV and DTP3 coverage maps, I would recommend to not make this comparison the primary focus of the paper. If a more formal, statistical analysis is possible with available data, then I would recommend presenting that analysis instead of the current description, to strengthen the paper with its current focus.

Response: Thank you for this comment. We have clarified the rationale behind the comparison between MCV and DTP3 to explore the effects of delivery mechanisms in our response to the General Comment. The additional data obtained from WHO stating when SIAs took place in the study countries only served to contextualize our comparison of the two vaccinations as we have now clarified in the paper.

The differences between MCV and DTP3 coverage have been quantified using additional statistics in the Results section: Comparing DTP3 with MCV coverage maps.

Comment #4 The authors discuss the difference in coverage between DTP3 and MCV, but did not quantify this in the results section. Would it be possible to show maps/statistics of the difference between DTP3 and MCV at the 1x1 km resolution and then identify areas where RI is clearly underperforming (stating the limitation that not all areas conducted SIA's)?

Response: Thank you for this comment. We showed maps of the differences between MCV and DTP3 coverage in the third column of Figure 4. We had used 50 km² hexagons to map the differences in order to reveal the patterns more clearly and as a standardized spatial scale to enable comparisons across the countries (the admin areas within these countries are not proportional in size and shape, hence not useful for our purpose here). In the current revision, we have also quantified the differences in percentages as: $(MCV - DTP3)/((MCV + DTP3)/2) \times 100$ to enhance our discussion; see Results Section. However, as requested here, the differences between MCV and DTP3 coverage at 1x1 km resolution are plotted as Supplementary Figure 13. At the country level, the differences between MCV and DTP3 coverage is further illustrated using a violin plot (see Supplementary Figure 12).

Our response to the General Comment precludes the need to state that not all areas conducted SIAs.

Comment #5 There was a substantial difference in model fit for DTP vs. MCV with much better fits for MCV, at least from R² values. Please explain this difference in model fit and what possible, unmodeled, factors could play a role in DTP but not in MCV?

Response: Thank you for this comment. The initial variation in model fit between MCV and DTP would have been partly produced by the different model specifications used in the analysis of the coverage of the two vaccinations in our initial draft. MCV was modelled in a univariate framework whereas DTP1-3 was modelled using a multivariate model. We had not used the predictive process model (see model-fitting details in supplemental materials) for MCV in some cases because the univariate models fitted for this antigen ran faster than the multivariate models used in the analysis of DTP. From our experience, fitting the full model, which could be impractical for some of the DTP analyses, instead of the predictive process model yields better predictive accuracy. For uniformity, we have specified the predictive process models for both vaccinations, using the same number of knot locations and prior specifications in each case. Interestingly, these additional steps taken to standardize both analyses have now yielded R² values and other validation statistics that are similar for the two vaccinations (see Table 1). We also note that any remaining differences in model performance could depend on the different sets of covariates included in the models for both vaccinations, the modelling framework (univariate vs multivariate) and how predictable the coverage of these vaccinations are by these covariates, among other factors.

Both the main manuscript and accompanying supplementary file have been edited and updated to reflect the changes made.

Comment #6 As far as I understand, DHS data are sampled to represent statistics at the regional level, which is a much coarser spatial resolution than the 1x1 km grid estimates used in this study. Why is it appropriate to make inference at the 1x1 km resolution when DHS data were sampled to represent a much higher resolution? And do the DHS data allow specific inference of spatial statistics at a higher resolution, e.g., as made in line 211:212 "The capital.... all three doses"?

Response: Thank you. DHS surveys are indeed designed to be representative at the regional (typically administrative level one areas) and national levels. However, the need for estimates of indicators at finer spatial scales to support program planning and implementation following the launch of the SDGs in 2015, and the increased availability of the Global Positioning System (GPS) coordinates of survey clusters have facilitated the use of DHS data for spatial interpolation. There are now many published works using model-based geostatistical techniques that leverage geospatial covariate data to produce subnational estimates of indicators measured during DHS surveys. In fact, the DHS program has now established its own research agenda in this area, having published many reports on the utilization of DHS data in the production of interpolated surfaces of indicators; see DHS Spatial Analysis Reports 9, 11, 14-16 (<https://dhsprogram.com/publications/publication-search.cfm?type=45>). They also routinely make available high resolution datasets using these approaches:
<https://spatialdata.dhsprogram.com/modeled-surfaces/>.

We have inserted a sentence in the methods section to address this.

Comment #7 The uncertainty maps for some countries, such as DRC, show a very distinct dotted pattern, with high uncertainty in some very specific places. What is the explanation for this pattern? And, assuming that the uncertainty levels are mostly related to the location of the DHS clusters (more uncertainty in areas without DHS clusters), could the authors also show a map of each country with the location of the clusters?

Response: Thank you for this comment. This is a very good point which we have now investigated and resolved by adjusting some of the model specifications. The patterns were caused by the numbers of knot locations used in the predictive process models that facilitate efficient computation (see modelling information in supplemental materials) and the extent of spatial correlation (spatial range) estimated through the spatial random effect in some cases. This problem was resolved by increasing the number of knot locations used in model-fitting and using more informative lower bounds in the Uniform prior distributions for the spatial range parameter in the fitted models. We further note that because of these changes, the dotted patterns no longer exist in the dropout rate maps as before – see Figure 2 and Supplementary Figures 7-10.

The maps of the locations of the DHS clusters are shown in Supplementary Figure 11. We did observe higher uncertainty in areas with fewer DHS clusters in some cases, see e.g. DTP maps for Cambodia, but this was not always the case. Uncertainty levels/patterns can also depend on other modelling considerations as highlighted above.

Comment #8 Table S1: It would help the interpretation of the models if the covariates could be described specifically, e.g., what exact poverty indicator, and the units of each covariate. Also, it is unclear why different variables were included for each country; what was the variable selection process?

Response: Thank you for this comment. Please see Supplementary Table 1A for a full description of all the covariates used in the analysis. We did describe the variable selection procedure briefly in the methods section, referencing our earlier paper which contained detailed information about this part of the analysis. However, we have edited the manuscript to highlight that we used the p-value approach which is relative to backward elimination for covariate selection. We have also mentioned that for each country, a uniform set of covariates was selected for DTP1-3 whereas a separate covariate combination was selected for MCV.

Minor comments

Comment #1 Introduction lines 57:59 "Understanding the spatial dynamics....programs": it will indeed be valuable to better understand spatial dynamics in complex environments with infectious disease transmission and various intervention programs, but this study did not evaluate dynamic, but rather static images of vaccination coverage. This study also separately analyzed each indicator (DTP3 and MCV), instead of a analysis of dynamic patterns of disease and both vaccination strategies (e.g., representing changes over time and interactions between vaccination strategies). It may be beyond the scope of this manuscript to study dynamic patterns, but given the statement on their importance in the introduction, the paper should probably mention the design/components of a study that would represent dynamic processes.

Response: Thank you. The word 'dynamics' was used here to mean 'fluctuation' or 'variation'; i.e. how vaccination coverage changes across space. To avoid any ambiguity, this word has been changed to 'variation'. We appreciate the comment on analysing the dynamic patterns of diseases vis-a-vis vaccination strategies as we intend to investigate this in future work. This has now been included in the discussion section, as pointed out by another reviewer.

Comment #2 Line 105 "strong association": consider replacement with "statistically significant association".

Response: This has been done. Thank you.

Comment #3 Lines 119-134 "The parameters...as expected": consider moving this section to the discussion, it distracts from the story and the stronger spatial dependence for DTP1 vs. 3 is explained as with the dropout process, which is not the main focus of the manuscript, but a secondary focus.

Response: Thank you for this comment. This paragraph contains the interpretation of the parameters of the spatial random effects included in the models for both MCV and DTP1-3. These parameters are an essential part of the modelling framework and should be interpreted/discussed alongside other modelling results. Besides, the paragraph has now been shortened to reflect the updated results.

Comment #4 Lines 137:141 "Table 1...Ethiopia": The bias should be summarized separately for MCV and DTP, since these are the programs under comparison.

Response: Thank you for this comment. We have edited these sentences to reflect the point made here.

Comment #5 Line 141 "VMSE": VMSE should be written in full at first mention in the text.

Response: Done. Thank you.

Comment #6 Lines 143-148 "For Cambodia.... 94.9%": This sentence should be written more clearly and precisely, e.g., it is not clear what "achieved coverages of 95% predication intervals were close to the true value of 95%" means. The following sentences inherit some of this lack of clarity. Maybe it would help to first introduce the concept of comparing 95% prediction intervals of model estimates with 95% confidence intervals of the observed data, or something similar, before listing the values for each country.

Response: This has been done as recommended. We have included a sentence clarifying the purpose of the achieved coverage of the 95% credible interval as measuring the accuracy of the uncertainties associated with the predictions.

Comment #7 Lines 307-309 "There are also biases.... underserved areas": It would be helpful to describe these biases in a bit more detail, particularly the direction in which these biases could have affected study results.

Response: Done as recommended. Thank you.

Comment #8 Lines 314-317 "Geographical.... educational status": The authors included various covariates in their vaccine coverage prediction models, such as remoteness or environmental factors, so these associations could shed light on what factors drive vaccination coverage. If other factors may exist that are statistically significantly associated with vaccination coverage, these would improve the spatial prediction models as well. In that context, the authors should explain why they selected the covariables currently included in the model, but left others for future work. I understand that no study can measure the effect of all possible covariates, but would it not make sense to explore major factors such as education, ethnicity, and e.g., nutritional status, for possible inclusion in prediction models of vaccination coverage? In that context, it would be helpful to describe the variable selection process in more detail in the methods section (see also previous comment about that).

Response: Thank you for this comment. The primary objective for the inclusion of covariates in the fitted models was to enhance the predictive ability of the models and not necessarily to determine

causal relationships between these and vaccination coverage. There are a wide range of factors that influence vaccination coverage. We could not include some of the additional variables mentioned in the Discussion in the current work due to the unavailability of their gridded surfaces – at least at the time of analysis. This part of the Discussion has now been edited to reflect these points and to point out that methodological approaches enabling the joint prediction of these covariates and vaccination coverage where relevant data exist at the survey cluster locations could be explored in future work.

We have also clarified the covariate selection process following major comment #8.

Comment #9 Lines 269-273 "Problems...infrastructure": Given that DTP doses are spaced very closely (within months of each other), it is unlikely that long-term processes such as conflicts, may be the main determinant of missing doses. Would it not be expected that the same factors explain low coverage and missing doses, such as remoteness, poverty, low education, etc.? A specific example contrasting determinants of completing a schedule vs. overall coverage, or additional citations, would strengthen this section.

Response: Thank you for this comment. We do agree that conflict is more likely to be associated with non-vaccination, but by conventional wisdom, it is also likely to lead to dropouts especially in areas with security challenges. We have now edited the manuscript to reflect this balanced view by mentioning it as a possible factor responsible for poorer coverage with latter doses in *areas with low DTP1 coverage*.

Our arguments in this part of the discussion were geared towards: (i) highlighting factors likely responsible for non-vaccination with any DTP dose (i.e. gaps in the coverage of DTP1) and (ii) pointing out that potential determinants of dropouts in areas with high DTP1 coverage may be different from causes of dropouts in areas with low DTP1 coverage. In response to this comment, we have included more factors and references to support these points. Although we did not set out to contrast the determinants of non-vaccination with those of dropouts, we have given an example from the literature, as suggested here, to highlight that the same factors/determinants could be responsible for the occurrence of both events.

Comment #10 Supplement line 43 "validation locations": Just to be sure, are these locations the DHS clusters or the 1x1 km grid cells? From the description, I would assume these would be the 1x1 km grid cells.

Response: Thank you for this comment. We have now clarified that the validation locations are DHS cluster locations both in this sentence and the one before it.

Response to comments by Reviewer #3

Comment #1 The reliance on DHS data should be made clear in the abstract and introduction, and perhaps also the caveat (already mentioned in the Discussion) that extrapolating to areas without DHS cluster data can be problematic when there are areas which cannot be surveyed (e.g. parts of northern Nigeria including Borno state).

Response: Thank you. The reliance on DHS data is now mentioned in both the Abstract and Introduction. We are of the opinion that the limitations of the data and methods are better suited for the Discussion where these have already been addressed.

Comment #2 It is not clear to me why a non-spatial model is used to perform model selection in terms of included covariates. Is this simply to save computation time?

Response: Thank you for this comment. This is a standard practice which is based on the notion that we want to explain as much of the variation in the data as possible using available covariates before accounting for any residual spatial autocorrelation. This has been mentioned in the Methods section.

Comment #3 There are opportunities to further validate these coverage maps aside from using cross-validation methods. This includes the use of alternative immunisation coverage data (e.g. WHO surveys, polio surveillance data). There are also opportunities to compare coverage with incidence of disease (e.g. measles). Have the authors plans to include such data? Could this be mentioned?

Response: Thank you for these insightful comments. We agree that the predicted maps presented in the paper could be validated using external data sources. The point about analysing vaccination coverage vis-a-vis disease incidence is also an important one. All of these have now been mentioned as part of future work in the Discussion section of the manuscript.

REVIEWERS' COMMENTS:

Reviewer #2 (Remarks to the Author):

I found the manuscript greatly improved in both clarity and content. Model fits seem to have improved after modifications in model fitting procedures, which was good to see. I have one additional comment, after reviewing the supplementary maps in the supplement. I don't have a good "feel" for the underlying data, given that most of the results section is devoted to presenting model results. The spatial distribution of uncertainty is driven by variability in the observations, the distribution of clusters, and model fitting procedures. I would like to see a simple histogram or other distribution plot of the underlying vaccination coverage rates, per country, if possible, to get a better feel for the consistency and variability of the data before the modeling process. This is a great paper and I recommend publication.

Response to reviewers' comments

Comments by Reviewer #2:

I found the manuscript greatly improved in both clarity and content. Model fits seem to have improved after modifications in model fitting procedures, which was good to see. I have one additional comment, after reviewing the supplementary maps in the supplement. I don't have a good "feel" for the underlying data, given that most of the results section is devoted to presenting model results. The spatial distribution of uncertainty is driven by variability in the observations, the distribution of clusters, and model fitting procedures. I would like to see a simple histogram or other distribution plot of the underlying vaccination coverage rates, per country, if possible, to get a better feel for the consistency and variability of the data before the modeling process. This is a great paper and I recommend publication.

Response: Thank you for these comments. The spatial distribution of the survey clusters as well as the corresponding observed vaccination coverage rates are now plotted in Supplementary Figure 11 for both MCV and DTP1-3 and all five study countries. This figure clearly shows the variability and other attributes of the modelled data sets.